# Influence of maternal use of tenofovir disoproxil fumarate or zidovudine in Vietnamese pregnant women with HIV on infant growth, renal function, and bone health

Ei Kinai[1,2]*, Hoai Dung Thi Nguyen[3], Ha Quan Do[4], Shoko Matsumoto[2], Moeko Nagai[2], Junko Tanuma[2], Kinh Van Nguyen[3], Thach Ngoc Pham[3], Shinichi Oka[2]

1 Department of Laboratory Medicine, Tokyo Medical University, Tokyo, Japan, 2 AIDS Clinical Center, National Center for Global Health and Medicine, Tokyo, Japan, 3 National Hospital for Tropical Diseases, Hanoi, Vietnam, 4 National Hospital of Obstetrics and Gynecology, Hanoi, Vietnam

* ekinai@tokyo-med.ac.jp

**Data Availability Statement:** All relevant data are within the paper and its Supporting Information file.

## Abstract

Tenofovir disoproxil fumarate (TDF) is still widely prescribed for human immunodeficiency virus (HIV)-infected pregnant women, despite its renal and bone toxicity. Although TDF-exposed infants often show transient growth impairment, it is not clear whether maternal TDF causes infantile rickets via maternal/fetal renal dysfunction in Asian populations. This prospective observational study was conducted in Vietnam and involved pregnant HIV-infected women treated with TDF-based regimen (TDF group) or zidovudine-based regimen (AZT-group). At birth, 3, 12, and 18 months of age, and included body length, weight, head circumference, serum alkaline phosphatase (ALP), creatinine, calcium, phosphorus, urine-β2-microglobulin (U-BMG), percentage of tubular reabsorption of phosphate (%TRP), and radiographic wrist score for rickets. Age-adjusted multivariate linear regression analysis evaluated the association of TDF/AZT use during pregnancy with fetal renal function and bone health. The study included 63 mother-infant pairs (TDF group = 53, AZT group = 10). In the mothers, detectable U-BMG (>252 μg/L) was observed more frequently in the TDF-than AZT group (89 vs 50%, p<0.001), but other renal/bone parameters were similar. In infants, maternal TDF use was not associated with growth impairment, renal dysfunction, or abnormal bone findings, but with a slightly higher ALP levels (p = 0.019). However, shorter length was associated with maternal AZT (p = 0.021), and worse radiographic scores were associated with LPV/r (p = 0.024). In Vietnamese population, TDF usage during pregnancy was not associated with infant transient rickets, growth impairment, or renal dysfunction, despite mild maternal tubular impairment. Maternal AZT and LPV/r influenced infant growth and bone health, though further studies are needed to confirm this finding.

**Funding:** This research was supported by the Japan Initiative for Global Research Network on Infectious Diseases from the Japan Agency for Medical Research and development, AMED. During the study period, antiretrovirals for Vietnamese patients were provided under the financial support of the U.S. president's emergency plan for AIDS relief (PEPFAR). The funders had no role in study design, data collection and analysis, decision to publish, or preparation of the manuscript.

**Competing interests:** E.K. has received honoraria from Gilead Sciences, ViiV Healthcare, and MSD. S. O. received honoraria from MSD, Janssen Pharmaceutical and Gilead Sciences, and research grants from MSD, ViiV Healthcare, Gilead Sciences. The remaining authors declare no conflict of interest. This research was supported by the Japan Initiative for Global Research Network on Infectious Diseases from the Japan Agency for Medical Research and development, AMED. During the study period, antiretrovirals for Vietnamese patients were provided under the financial support of the U.S. President's Emergency Plan for AIDS Relief (PEPFAR). This does not alter our adherence to PLOS ONE policies on sharing data and materials.

## Introduction

Tenofovir disoproxil fumarate (TDF) is still widely used for treatment of human immunodeficiency virus (HIV) infection. In the past two decades, zidovudine (AZT)-based treatment was the only approved antiretrovirals for pregnant women and infants, despite its lower potency and toxicity. In 2013, the World Health Organization (WHO) recommended the use of TDF-based two nucleoside reverse transcriptase inhibitors (NRTIs) for pregnant women based on the high potency [1]. However, TDF is known to be nephrotoxic [2], mainly causing proximal tubular dysfunction [3, 4], which results in renal loss of phosphorus with a subsequent decrease in bone mineral density (BMD) in both the lumbar spine and pelvic bones [5, 6]. This mechanism was confirmed in a study that showed tenofovir alafenamide (TAF), a novel pro-drug, reduced TDF-related nephrotoxicity, resulting in significant improvement in not only renal tubular function but also in bone mineral density [3].

Based on the toxic effects of TDF on renal function and bone health, the safety of TDF in pregnant woman is a matter of concern. Especially given that TDF use during pregnancy can theoretically cause nutritional rickets during early neonatal life due to maternal renal loss of phosphate in late pregnancy [7]. Previous studies in rhesus macaques showed that TDF administration in neonates and infants resulted in severe rickets [8]. In human, most of the large international studies suggest that maternal TDF was not associated with infant growth impairment [9–11]. However, a few longitudinal cohort studies reported growth retardation of TDF-exposed infants within a few months after birth, though this was described as transient and self-corrected by 1–2 year of age [9, 12]. In addition, while whole-body bone mineral content was lower in TDF-exposed infants at the first 4 weeks after birth [13], lumbar spine BMD showed comparable level at 12 months after birth [14]. These findings suggest that maternal TDF can cause transient nutritional rickets due to maternal renal loss of phosphates, which can be overcome by nutritional replacement after birth. Furthermore, there is limited information on the effects of maternal TDF on infant growth in Asian population [14, 15], despite being considered vulnerable to TDF-associated nephrotoxicity due to the small body size [16].

The present study was designed to evaluate whether maternal TDF can cause transient nutritional rickets by impairing infant growth, renal function, or bone health in Vietnamese pregnant women with HIV.

## Materials and methods

### Sample population

This prospective observational study started in December 2016 and completed in December 2019. Vietnamese HIV-infected pregnant women who were already on anti-retroviral therapy (ART) were recruited at the National Hospital of Obstetrics and Gynecology (NHOG) before the third trimester. Each participant provided a signed informed consent after a thorough explanation of the purpose of the study and potential outcome. The inclusion criteria were as follows; 1) HIV-infected pregnant women, 2) free of acute AIDS-defining diseases, and 3) on treatment at the time of study entry with either TDF- or zidovudine (AZT)-based regimen, either after 26 gestational weeks or for more than 8 weeks during pregnancy. We also applied the following exclusion criteria; 1) HIV-infected infants (confirmed by HIV antibody test at 18 month of age), 2) preterm infants (<35 gestational weeks or <2000 g of birth weight), 3) multiple pregnancy (twins, triplets, or more), and 4) congenital disorders (e.g., failure to thrive, rickets or renal impairment). Since sex-specific standard normal values for height, weight, head circumference and laboratory values are not available for Vietnamese newborn/infants, we also recruited HIV-non-infected mothers and their infants in this study.

The study was approved by the Institutional Review Board of the National Center for Global Health and Medicine (NCGM), Tokyo, Japan (#NCGM-G-002030-00), National Hospital of Tropical Diseases (NHTD), Hanoi, Vietnam (#18/HDDD-NDTU), and National Hospital of Obstetrics and Gynecology (NHOG), Hanoi, Vietnam (#936/CN-PSTW).

## Data collection

The enrolled pregnant women were asked to visit the NHTD at 32–36 gestational week for blood and urine tests on the viral load, cluster of differentiation 4 (CD4) cell count, and renal function (Fig 1). The NHOG provided delivery care support, checked whether newborns met the inclusion criteria, measured the body length, weight, and head circumference at birth, and recorded baseline clinical information, such as the Apgar score. The latter comprises five items related to the status of the newborn, which are measured immediately after birth: skin color, heart rate, tendon reflexes, muscle tone, and respiration. After the delivery, the mother-infant

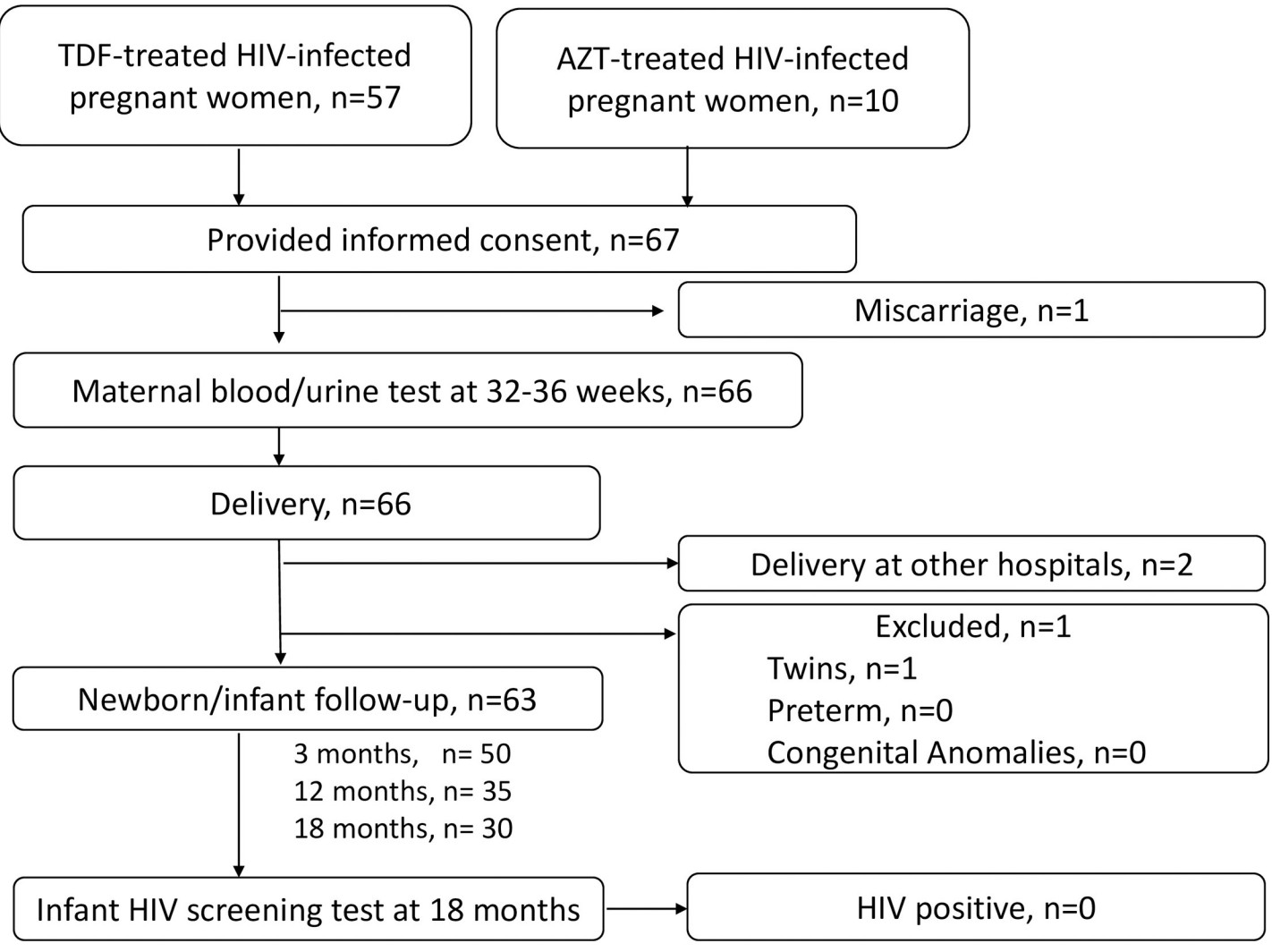

**Fig 1. Flow chart of the subject enrollment process.** Sixty-seven HIV-infected and 10 HIV-negative pregnant women were recruited at the NHOG. Of the 57 TDF-treated mother-infant pairs, 4 pairs were excluded due to miscarriage (n = 1), delivery at other hospitals (n = 2), and twins (n = 1). Although 3 of the 10 HIV-negative mothers were enrolled and provided maternal data and infant physical data at birth, none of the infants was brought to the center for further physical measurement and laboratory tests.

pairs, who conformed the study inclusion and exclusion criteria, visited the NHTD at 3, 12, and 18 months after delivery for maternal and infantile measurements of various physical parameters and laboratory tests. Specifically, the infant length, weight and head circumference, and serum alkaline phosphatase (ALP), creatinine, calcium, phosphorus, urine-β2-microglobulin (U-BMG), percentage of tubular reabsorption of phosphate (%TRP), and wrist radiographic score for rickets were obtained at these three visits. U-BMG was measured using a turbidimetric immunoassay with a coefficient of variation (CV) of 0.9–2.9% in manufacturer's data (Abbott Quantia β2-Microglobulin reagent kit with ARCHITECT ci8200, Abbott Diagnostics, Abbott Park, IL). The %TRP was calculated using urine-phosphate and urine-creatinine measured from the spot urine sample, and serum-creatinine and serum-phosphorus levels obtained from blood samples drawn on the same day. Serum/urine phosphate and creatinine were measured using kinetic color test (Beckman Coulter AU480, Beckman Coulter, Brea, CA). At 3 months of age, serum 25 (OH) Vitamin D level was determined at NHTD using a delayed 1-step chemiluminescence microparticle immunoassay (CMIA) (Abbott ARCHITECT 25-OH Vitamin D assay reagent kit with ARCHITECT i2000SR immunoassay analyzer, Abbott Diagnostics, Abbott Park, IL). Its CV and the mean bias to the reference method were reported 2.7–4.6% and -16.3%, respectively [17].

## Radiographic scoring for nutritional rickets

Infant wrist radiographic score for nutritional rickets was determined at 3, 12, and 18 months of age, using a widely adapted 10-point scoring system [18]. Briefly, a grade of 0 represents a normal wrist, a grade 1 represents a widened growth plate and irregularity of metaphyseal margin without concave cupping, and a grade 2 represents metaphyseal concavity with fraying of margins. The maximum score is 4, with 2 for both the radius and ulna. Each radiograph was scored independently by a trained physician at the National Center for Global Health and Medicine, who was blinded to clinical information.

## Statistical analysis

The clinical characteristics of the mothers and the infant baseline data were compared between the TDF and AZT groups using the chi-square test for nominal data and by the Mann-Whitney test for continuous parameters. The infant growth outcomes (length, weight, and head circumference) of the TDF and AZT groups were compared by two-way ANOVA at different ages. The prevalence of detectable U-BMG (>252 μg/L) and abnormal radiographic score for rickets in the two treatment groups were compared by the chi-square test. We also used linear multiple regression analyses to determine the relationship between growth parameters and the laboratory test values, and confounding factors other than maternal use of TDF and infant age, where maternal use of LPV/r, maternal age, CD4, and viral load, and gestational age, and infant age were set as the independent variables, and the estimates, 95% confidence interval (95% CI) were calculated. For U-BMG, multivariate logistic analysis was conducted where the presence of detectable level of U-BMG was set as the dependent variable with the same independent variables as in multiple regression analyses. All statistical analyses were performed using The Statistical Package for Social Sciences ver. 23.0 (SPSS, Chicago, IL).

## Results

### Sample population

A total of 67 HIV-infected pregnant women (57 TDF-treated women and 10 AZT-treated women) were enrolled in this study before the third trimester. Of those, 63 mother-infant

pairs (TDF group, n = 53, AZT group, n = 10) who delivered at the NHOG met the study criteria (Fig 1). Although more than 30 non-HIV mothers were recruited, only 2 mother-infant pairs agreed to participate in the present study. While both pairs provided maternal demographic and laboratory data and infant physical data at birth, none visited our facility for further infant follow-up. The median maternal age of the 63 mothers was 32 years [interquartile range (IQR) 8.0], and median body weight during 32–36 gestational weeks was 57.5 kg (IQR 11.0), suggesting a non-pregnant body weight of less than 50 kg, similar to the majority of Vietnamese women (Table 1). There were no significant differences in age, body weight, VL and CD4 cell count between the 53 TDF and 10 AZT groups. Based on the 2013 WHO guidelines, which recommended the combination of TDF+3TC+EFV for all HIV-infected patients,

**Table 1. Clinical characteristics and laboratory test results of pregnant women at 32–36 weeks of gestation.**

| | Total (n = 63) | TDF (n = 53) | AZT (n = 10) | P value |
|---|---|---|---|---|
| Age (years), median (IQR) | 32 (8.0) | 32 (6.1) | 32 (8.7) | 0.805 |
| Body weight (kg), median (IQR) | 57.3 (11.0) | 57.5 (12.3) | 57.3 (6.5) | 0.601 |
| Gestational complications, n (%) | 2 (3.2) | 1 (1.9) | 1 (10) | 0.382 |
| Current Smoking, n (%) | 1 (1.6) | 1 (1.9) | 0 (0) | n.a. |
| Alcohol use, n (%) | 0 (0) | 0 (0) | 0 (0) | n.a. |
| CD4 cell count (median, IQR) | 298 (201) | 394 (191) | 442 (205) | 0.689 |
| HIV-RNA undetectable (<20) (n, %) | 49 (78) | 41 (77) | 8 (80) | 0.854 |
| Alkaline phosphatase (IU/L), median (IQR) | 102 (33) | 104 (33) | 84 (40) | 0.162 |
| Creatinine (mg/dL), median (IQR) | 0.59 (0.16) | 0.59 (0.15) | 0.53 (0.19) | 0.276 |
| Calcium (mg/dL), median (IQR) | 8.8 (0.5) | 8.7 (0.4) | 8.8 (0.4) | 0.316 |
| Phosphorus (mg/dL), median (IQR) | 3.3 (0.6) | 3.1 (0.9) | 3.6 (0.5) | 0.358 |
| U = BMG (μg/L), median (IQR) | 2755 (5222) | 3060 (5915) | 581 (3322) | <0.001 |
| Detectable U-BMG (>252 μg/L), n (%) | 52 (83) | 47 (89) | 5 (50) | <0.001 |
| %TRP (%), median (IQR) | 92.2 (6.2) | 91.5 (6.1) | 93.8 (6.1) | 0.241 |
| Low %TRP (<90%), n (%) | 22 (34.9) | 20 (38.5) | 2 (20.0) | 0.309 |
| Very low %TRP (<80%), n (%) | 0 (0) | 0 (0) | 0 (0) | n.a. |
| *Hemoglobin (g/dL), median (IQR) | 12.4 (1.4) | 12.6 (1.7) | 12.1 (0.8) | 0.260 |
| *Hematocrit (%), median (IQR) | 36.9 (4.5) | 37.3 (4.6) | 35.3 (1.1) | 0.214 |
| *Mean corpuscular volume (fL), median (IQR) | 97.4 (11.2) | 95.7 (7.2) | 109.3 (20.5) | 0.051 |
| History of AIDS-defining illness, n (%) | 0 (0) | 0 (0) | 0 (0) | n.a. |
| ART initiation | | | | |
| Before pregnancy, n (%) | 56 (89) | 47 (89) | 9 (90) | 0.903 |
| During pregnancy, n (%) | 7 (11) | 6 (11) | 1 (10) | |
| ART duration (months), median (IQR) | 29 (51.7)) | 28 (55.3) | 47 (53.8) | 0.155 |
| History of TDF use, n (%) | 57 (90) | 53 (100) | 4 (40) | n.a. |
| TDF duration (months), median (IQR) | 22 (38.3) | 26 (48.2) | 3 (28.3) | 0.235 |
| Key medications during pregnancy | | | | |
| EFV, n (%) | 49 (77.8) | 49 (92.4) | 0 (0) | n.a. |
| NVP, n (%) | 7 (11.1) | 2 (3.8) | 5 (50) | <0.001 |
| LPV/r, n (%) | 7 (11.1) | 2 (3.8) | 5 (50) | <0.001 |

P value was determined by chi-square test of t-test in comparison of TDF and AZT group.

*Blood samples for hemoglobin, hematocrit, and mean corpuscular volume were collected from 31/63 participants (23 of TDF-treated mothers and 8 of AZT-treated mothers) for post-hoc analysis.

IQR: interquartile range, TDF; tenofovir disoproxil fumarate, AZT; zidovudine, U-BMG; urine-β2-microglobulin, %TRP; the percentage of renal tubular reabsorption of phosphate, EFV; efavirenz, NVP; nevirapine, LPV/r; lopinavir-ritonavir (Kaletra)

including pregnant women, 53 of the 63 (84%) pregnant women received TDF-based regimen, while 49 of the 63 (78%) received TDF+3TC+EFV. Furthermore, LPV/r was used by 5 of the 10 (50%) mothers of the AZT group, but only by 7 of the 53 (11.1%) of the TDF group.

## Effects of TDF on maternal laboratory tests

Laboratory tests showed a tendency for higher levels of serum creatinine (0.59 vs. 0.53 mg/dL, p = 0.276), lower concentrations of serum phosphorus (3.1 vs. 3.6 mg/dL, p = 0.358), and higher percentage of mothers with low %TRP (38.7% vs. 20.0%, p = 0.309) in the TDF group, compared with the AZT group. Furthermore, a significantly larger proportion of mothers of the TDF group had high U-BMG compared with the AZT group (89% vs. 50%, p<0.001) (Table 1).

## Effects of TDF and AZT on newborn physical characteristics

The majority of infants were born at 39 weeks of gestation and none showed perinatal complications or congenital abnormalities. Body length and head circumference at birth were smaller in the AZT group compared with the TDF group [49.2 vs. 46.4 cm, p = 0.061 (body length), and 34.0 vs. 32.5 cm, p<0.001 (head circumference) in TDF and AZT group, respectively]. In Vietnam, it is the Government policy to treat all infants of HIV-infected mothers with nevirapine, but not AZT, during the first 6 weeks of life. The 25 (OH) vitamin D level at 3 months of the entire group was low (median: 32.1 IU/mL, IQR 13.6 IU/mL), and 24 of the total 63 (38.1%) infants had low levels of vitamin D (Table 2).

## Comparative analysis of effects of TDF and AZT on infant growth, renal function and bone health

Newborn body length, weight, and head circumference were significantly smaller in the AZT than TDF group (2-way ANOVA, adjusted for age (p = 0.002, 0.047, and 0.049, respectively)

**Table 2. Baseline clinical characteristics of the study infants.**

| | Total (n = 63) | TDF (n = 53) | AZT (n = 10) | P value |
|---|---|---|---|---|
| Sex (male): n, (%) | 37 (58.7) | 34 (64.2) | 3 (30) | 0.129 |
| Gestational age (week), median (IQR) | 39 (1.1) | 39 (1) | 39 (1.3) | 0.462 |
| Birth length (cm), median (IQR) | 48.0 (2.3) | 49.2 (1.7) | 46.0 (8.5) | 0.061 |
| Birth weight (g), median (IQR) | 3100 (500) | 3200 (550) | 2900 (750) | 0.165 |
| Birth head circumference (cm), median (IQR) | 34.0 (2.0) | 34 (1.8) | 32.5 (5.6) | <0.001 |
| APGAR score (1 min), median (IQR) | 9 (1) | 9 (1) | 9 (1) | NS |
| APGAR score (5 min), median (IQR) | 10 (0) | 10 (0) | 10 (0) | NS |
| Perinatal complication, n (%) | 0 (0) | 0 (0) | 0 (0) | NS |
| Congenital abnormalities, n (%) | 0 (0) | 0 (0) | 0 (0) | NS |
| Nutrition (breastfeeding), n (%) | 0 (0) | 0 (0) | 0 (0) | NS |
| Postpartum antiretroviral | | | | |
| Zidovudine, n (%) | 0 (0) | 0 (0) | 0 (0) | NS |
| Nevirapine, n (%) | 63 (100) | 53 (100) | 10 (100) | NS |
| Duration of postpartum antiretrovirals (wks), median (IQR) | 6 (0) | 6 (0) | 6 (0) | NS |
| 25-OH Vitamin D at 3 months (IU/mL), median (IQR) | 32.1 (13.6) | 32.3 (13.8) | 27.6 (15.6) | 0.411 |
| Low 25-OH Vitamin D at 3 months (<30 IU/mL), n (%) | 24 (38.1) | 19 (35.8) | 5 (50) | 0.428 |
| Deficient (<20 IU/mL), n (%) | 6 (9.5) | 6 (11.3) | 0 (0) | 0.316 |
| Insufficient (20–30 IU/mL), n (%) | 18 (28.6) | 13 (24.5) | 5 (50) | 0.242 |

IQR: interquartile range, TDF; tenofovir disoproxil fumarate, AZT; zidovudine, n.a.: not available, NS: not significant

(Fig 2(A)–2(C)). Furthermore, ALP was significantly higher in the TDF compared with the AZT group (p = 0.007), whereas serum-creatinine, phosphorus and %TRP were not different (2-way ANOVA, p = 0.505, 0.583, and 0.972, respectively) (Fig 2(D)–2(G)).

Serial analysis showed a decrease in U-BMG level with age in both treatment groups, but no significant difference in the prevalence of detectable U-BMG between the TDF and AZT groups [5/53 (9.4%) vs. 2/10 (20%) at 3 months (p = 0.293), respectively (Fig 3A)]. There were no significant differences in the wrist radiographic scores for rickets between the two groups irrespective of age. Rapid improvement was observed with time after birth in the TDF group, and no abnormal findings being observed at 18 months of age. In comparison, one of the 5 infants of the AZT group (20%) continued to have poor radiographic scores as they aged and the score was still abnormal at 18 months of age (Fig 3B).

## Effects of confounding factors on infant growth, renal function and bone health

Multivariate linear regression analysis showed significant association between maternal TDF use and longer body length (estimate 2.52; 95% confidence interval (CI) 0.38–4.66, p = 0.021). Apart from infant age, no other confounding factors were associated with body length, weight, and head circumference (Table 3). Multivariate linear regression analyses with the same independent variables as those used in the models for physical parameters (Table 4) showed a significant association between high serum alkaline phosphatase levels and maternal TDF [estimates 41.22; 95%CI: 6.98–75.45), p = 0.019]. Overall, maternal TDF neither correlated with elevated serum creatinine, serum phosphorus, %TRP, radiographic score for rickets, nor with abnormal U-BMG. Interestingly, poor wrist radiographic score for rickets correlated significantly with maternal use of lopinavir/ritonavir (LPV/r), but not TDF (estimate 0.55; 95% CI: 0.08–1.02, p = 0.024).

## Post-hoc evaluation

Hemoglobin, hematocrit, erythrocyte count, and mean corpuscular volume (MCV) were measured during pregnancy in the HIV-infected mothers managed clinically at NHTD. Hemoglobin and hematocrit levels (measured in 31 mothers, TDF group: 23/53, AZT group: 8/10) were not significantly different between the AZT and TDF groups, while MCV was larger in the AZT than the TDF group (109.1 vs 95.7 fL, respectively, p = 0.051, Table 1).

## Discussion

The main finding of the present study was that TDF use by pregnant Vietnamese women was not associated with infant growth impediment, renal dysfunction, or nutritional rickets, although it correlated with mild maternal renal tubular dysfunction. While we did not investigate the mechanisms of these changes in the present study, we postulate the following mechanisms. First, since growth is rapid in both late pregnancy and early infancy [7, 19], preterm infants are at risk of metabolic bone diseases in the presence of low supplies of phosphorus and calcium [20]. Similarly, TDF-associated maternal renal tubular dysfunction during late pregnancy can cause blunted fetal growth based on inadequate supply of phosphate [21]. In the present study, although TDF use during pregnancy was associated with significantly high maternal U-BMG levels, the maternal and infant %TRP levels were not significantly different between the TDF and AZT groups. These results suggest that TDF use by the Vietnamese pregnant mothers was not associated with significant renal loss of phosphorus. Since high serum levels of ALP in the infant were associated with maternal TDF use, consistent with the findings in adult populations [4, 22], maternal TDF exposure seems to influence bone health of infants

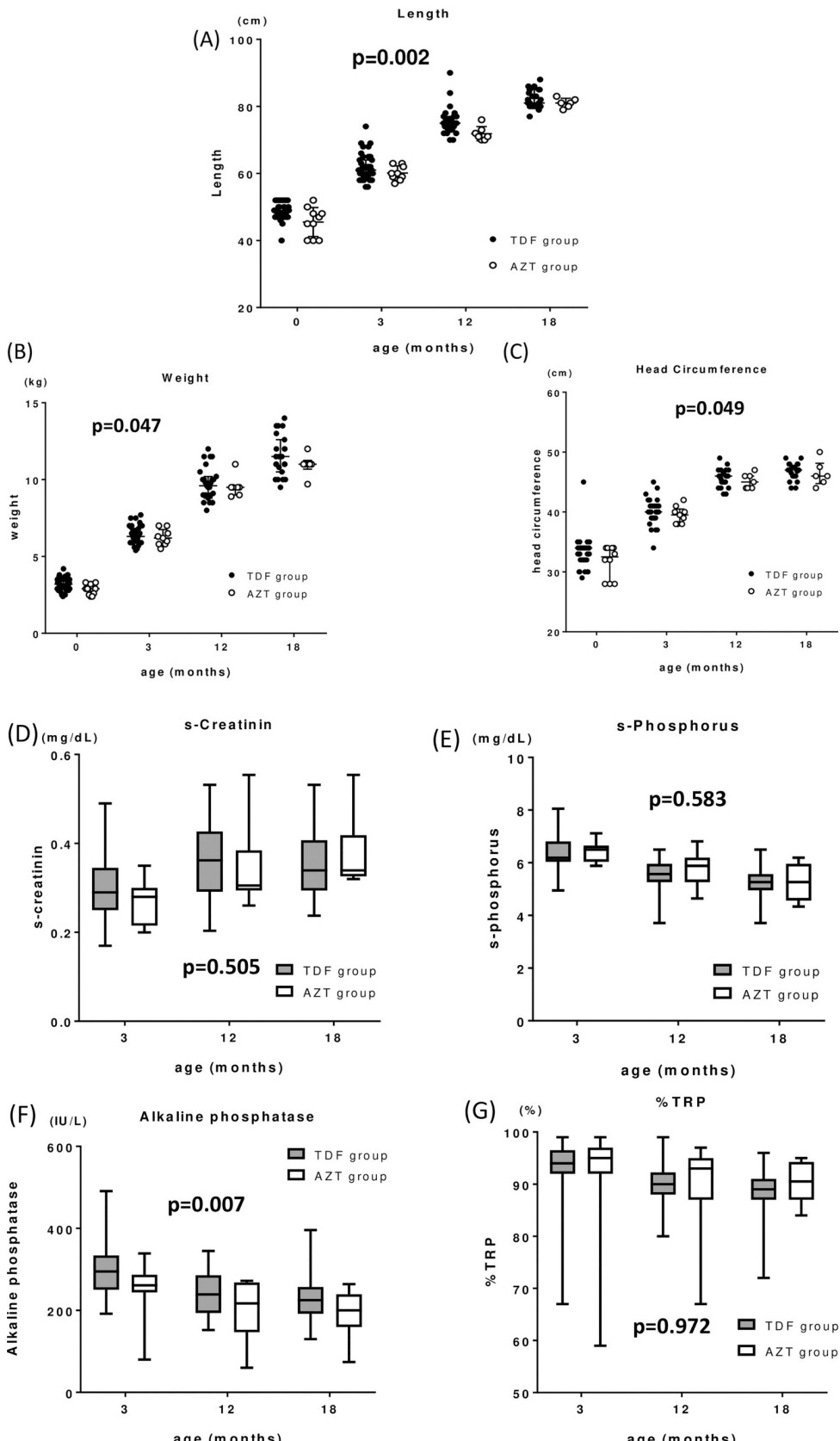

**Fig 2. The dot plots provide comparisons of neonate/infant length (A), weight (B), and head circumference (C) between the TDF and AZT groups.** The box-whisker plots provide comparisons of neonate/infant serum creatinine (D), serum phosphorus (E), serum alkaline phosphatase (F), and the percentage of tubular reabsorption of phosphate (G) between the two groups. Data were compared by 2-way ANOVA to adjust for the influence of age.

at subclinical levels. Second, infants aged 3 months had below normal levels of 25-OH Vitamin D (<30 IU/mL) (in 38% of TDF group and 50% of AZT group). These data are consistent with a number of previous studies that demonstrated higher rates of vitamin D deficiency among Asians, compared with Caucasians, probably due to insufficient dietary intake [23, 24]. Given vitamin D deficiency rickets is the most common etiology of rickets [7] especially in Asian countries, the lower serum levels of 25-OH vitamin D seem to mask the impact of TDF-associated infant growth disturbance via renal loss of phosphate.

Surprisingly, our study identified AZT, rather than TDF, as the only significant risk factor for shorter infant body length. Various AZT toxic effects have been reported, including mitochondrial dysfunction with myopathy and cardiomyopathy [25], and bone marrow suppression [26]. Although reviews of ACTG 076 supported the safety of *in utero* exposure to AZT with respect to long-term infant development/growth [27], serious concerns have been raised about infant mitochondrial dysfunction [28, 29] and cardiac toxicity [30]. Since AZT passes easily through the placenta, it confers high plasma concentrations in cord blood/peripheral circulation of infants [31]. Asian population with small body size could be vulnerable to AZT toxicity if exposed to AZT *in utero*. Unfortunately, blood samples were not obtained from the infants in the present study for ethical reasons and thus no hematological parameters, lactate levels, blood gases, or creatine kinase levels were measured to evaluate AZT toxicity in infants. Post-hoc analysis did not show any differences in hematologic parameters between TDF and

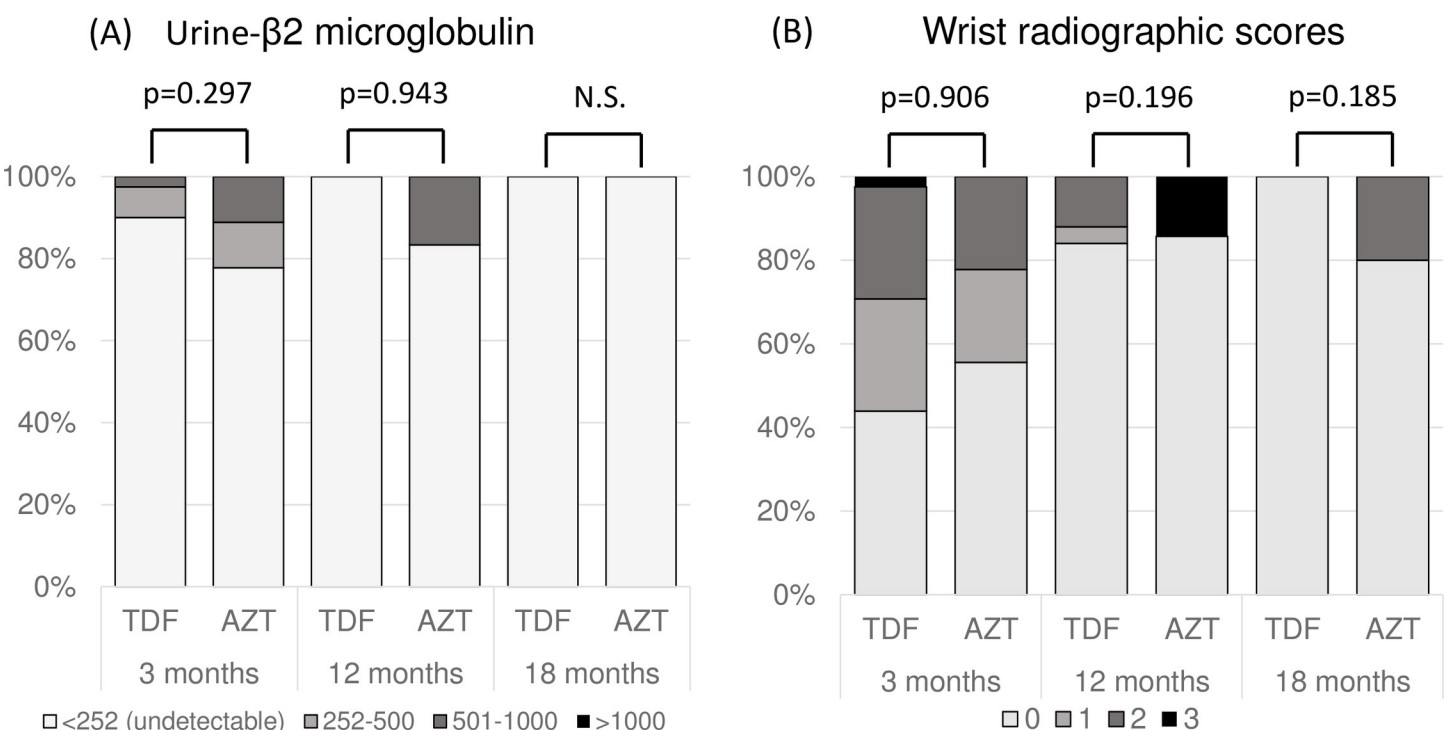

**Fig 3. Comparison of the prevalence of elevated urine β2-microglobulin (A) and abnormal radiographic wrist score for rickets (B) between the TDF and AZT groups.** Data were compared by chi-square test.

**Table 3. Results of multivariate regression analysis of the association of various maternal/infant factors with infant body length, weight, and head circumference.**

| | Length | | Weight | | Head Circumference | |
|---|---|---|---|---|---|---|
| | Estimate (95% CI) | P value | Estimate (95% CI) | P value | Estimate (95% CI) | P value |
| Maternal TDF (yes or no) | **2.52 (0.38 to 4.66)** | **0.021** | 130.1 (-393.1 to 653.3) | 0.624 | 0.47 (-0.81 to 1.74) | 0.470 |
| Maternal LPV/r (yes or no) | 0.61 (-1.84 to 3.06) | 0.626 | -75.6 (-675.4 to 524.2) | 0.804 | -0.34 (-1.79 to 1.12) | 0.643 |
| Gestational week at birth (per 1 week increase) | 0.20 (-1.69 to 1.35) | 0.542 | 33.2 (-127.0 to 193.4) | 0.683 | -0.16 (-0.55 to 0.24) | 0.434 |
| Infant sex (male = 1, female = 0) | -0.17 (-0.004 to 0.007) | 0.825 | -28.4 (-398.0 to 341.1) | 0.879 | -0.54 (-1.45 to 0.36) | 0.238 |
| Infant age (per 1 month increase) | 1.83 (1.73 to 1.93) | <0.001 | -149.0 (-174.7 to -123.3) | <0.001 | 0.75 (0.68 to 0.81) | <0.001 |

AZT maternal groups. Further studies are needed to determine the effects of AZT and TDF use during pregnancy on infant growth.

LPV/r was used in 5 of the 10 (50%) AZT-treated women but only in 2 of the 53 (3.8%) pregnant mothers of the TDF group. A number of well-designed randomized controlled studies reported significant association between maternal use of protease inhibitors, such as LPV/r, and fetal prematurity [32, 33]. A more recent study concluded that the use of LPV/r was the most significant determinant of neonatal delayed growth [34]. Especially, protease inhibitors are reported to have adverse effects on bone health [6, 35] through the activation of osteoclasts [36] or blockade of osteoblast differentiation [37]. Since bone ALP is the main isozyme in infants, the low levels of ALP observed in our study in the AZT group may partly explain the low osteoblast activity in LPV/r-exposed infants. Furthermore, given that perinatal and post-natal exposure to LPV/r, a potent cytochrome p450 inhibitor, causes clinically-evident adrenal

**Table 4. Results of multivariate regression analysis of the association of various laboratory findings with maternal and infant parameters.**

| | Maternal TDF (yes/no) | Maternal LPV/r (yes/no) | Gestational age (/week increase) | Infant sex (m = 1, f = 0) | Infant age (/month increase) |
|---|---|---|---|---|---|
| Serum creatinine | | | | | |
| Estimate (95% CI) | 0.03 (-0.02 to 0.07) | 0.03 (-0.03 to 0.08) | 0.009 (-0.005 to 0.023) | 0.01 (-0.02 to 0.04) | **0.005 (0.003 to 0.008)** |
| P value | 0.211 | 0.315 | 0.225 | 0.541 | **<0.001** |
| Serum phosphorus | | | | | |
| Estimate (95% CI) | -0.05 (-0.39 to 0.29) | -0.13 (-0.53 to 0.26) | -0.04 (-0.15 to 0.07) | 0.10 (-0.10 to 0.36) | **-0.08 (-0.10 to -0.06)** |
| P value | 0.773 | 0.507 | 0.474 | 0.427 | **<0.001** |
| Serum alkaline phosphatase | | | | | |
| Estimate (95% CI) | **41.22 (6.98 to 75.45)** | -14.42 (-54.01 to 25.17) | 7.92 (-3.04 to 18.87) | 5.64 (-20.02 to 31.3) | **-5.34 (-7.25 to -3.42)** |
| P value | **0.019** | 0.472 | 0.155 | 0.664 | **<0.001** |
| %TRP | | | | | |
| Estimate (95% CI) | 0.09 (-4.93 to 5.12) | 1.05 (-4.77 to 6.86) | **-1.68 (-3.29 to -0.07)** | -0.71 (-4.48 to 3.06) | **-0.33 (-0.61 to -0.53)** |
| P value | 0.971 | 0.722 | **0.041** | 0.709 | **0.020** |
| Wrist radiographic scores | | | | | |
| Estimate (95% CI) | 0.15 (-0.26 to 0.57) | **0.55 (0.08 to 1.02)** | 0.08 (-0.05 to 0.21) | -0.06 (-0.37 to 0.25) | **-0.05 (-0.08 to -0.03)** |
| P value | 0.463 | **0.024** | 0.242 | 0.709 | **<0.001** |
| Detectable U-BMG | | | | | |
| Odds ratio (95% CI) | 0.56 (0.11 to 2.80) | 0.37 (0.04 to 3.92) | 0.96 (0.90 to 1.03) | 1.21 (0.35 to 4.13) | 0.96 (0.87 to 1.06) |
| P value | 0.480 | 0.407 | 0.281 | 0.763 | 0.391 |

TDF: tenofovir disoproxil fumarate, LPV/r: the combination of lopinavir and ritonavir (Kaletra)

Differences in U-BMG were tested by logistic multivariate analysis, with U-BMG detectable level (>252 μg/L) set as the independent variable.

dysfunction [38], maternal exposure to LPV/r may have an adverse impact on infant growth or bone health.

Our study has certain limitations. First, the smaller sample size of the AZT group limited the statistical power for appropriate comparison of renal/bone health markers, because Vietnamese pregnant women had been treated mainly with TDF-containing regimens in accordance with the 2013 launched WHO recommendation [1]. This limitation also curtailed full analysis of the association of AZT and LPV/r with infant growth and bone health, although the main finding of the present study was the adverse effects of AZT and LPV/r. Second, infant growth should be compared using Z-scores to standardize the data for sex and race. To our knowledge, there are no established standard norms for physical growth of Vietnamese newborns and infants. In addition, normal values of tests of renal function, bone health, and serum levels of vitamin D are also not available. Although we tried to collect data of non-HIV mother-infant pairs to deal with the lack of normative data for Vietnamese, only two of the enrolled non-HIV mother-infant pairs visited our center at the time of the study. In the present study, the higher percentage of male infants in the TDF group (34 of 53; 64%), compared with the AZT group (3 of 10; 30%), may explain the significant difference in body length between the two groups. However, in general, the difference in body length between the sexes is only 1 cm from birth to 18 months of age [39], whereas the present study showed a larger than expected difference (49.2 vs. 46.0 cm at birth and 75.0 vs. 71.0cm at 18 months of age in the TDF and AZT groups, respectively). Moreover, multivariate linear regression analysis showed that female sex was not associated with shorter length (p = 0.825).

In conclusion, we have demonstrated in the present study that the use of TDF during pregnancy was not associated with harmful effects on neonatal and infant growth, renal function, or bone health, though it was associated with mild maternal renal tubular dysfunction in Vietnamese population. Maternal AZT and LPV/r showed significant association with infant growth and bone health, which requires further investigation.

## Supporting information

**S1 Dataset.**
(XLSX)

## Acknowledgments

We thank the children, caretakers and staff of all facilities who participated in the present study, especially M. Sata and H. Nguyen (AIDS Clinical Center), D.T. Thuy and P.T. Tuyen (NHTD, Hanoi, Vietnam), and N.T. Trang (NHOG, Hanoi, Vietnam).

## Author Contributions

**Conceptualization:** Ei Kinai.

**Data curation:** Hoai Dung Thi Nguyen, Ha Quan Do, Shoko Matsumoto, Moeko Nagai.

**Formal analysis:** Ei Kinai, Shoko Matsumoto.

**Funding acquisition:** Ei Kinai, Junko Tanuma, Shinichi Oka.

**Investigation:** Hoai Dung Thi Nguyen, Ha Quan Do, Shoko Matsumoto, Moeko Nagai.

**Project administration:** Ei Kinai, Hoai Dung Thi Nguyen, Ha Quan Do.

**Supervision:** Junko Tanuma, Kinh Van Nguyen, Thach Ngoc Pham, Shinichi Oka.

**Writing – original draft:** Ei Kinai.

**Writing – review & editing:** Shinichi Oka.

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
