## [Decision Letter · Decision Letter 0]

15 Mar 2021

PONE-D-21-05310

Influence of maternal use of tenofovir disoproxil fumarate or zidovudine in Vietnamese pregnant women with HIV on infant growth, renal function, and bone health

PLOS ONE

Dear Dr. Kinai,

Thank you for submitting your manuscript to PLOS ONE. After careful consideration, we feel that it has merit but does not fully meet PLOS ONE’s publication criteria as it currently stands. Therefore, we invite you to submit a revised version of the manuscript that addresses the points raised during the review process.

We look forward to receiving your revised manuscript.

Kind regards,

Linglin Xie

Academic Editor

PLOS ONE

Journal Requirements:

"E.K. has received honoraria from Gilead Sciences, ViiV Healthcare, and MSD. S.O. received honoraria from MSD, Janssen Pharmaceutical and Gilead Sciences, and research grants from MSD, ViiV Healthcare, Gilead Sciences. The remaining authors declare no conflict of interest.

This research was supported by the Japan Initiative for Global Research Network on Infectious Diseases from the Japan Agency for Medical Research and development, AMED. During the study period, antiretrovirals for Vietnamese patients were provided under the financial support of the U.S. president’s emergency plan for AIDS relief (PEPFAR)."

Reviewers' comments:

Reviewer's Responses to Questions

**Comments to the Author**

1. Is the manuscript technically sound, and do the data support the conclusions?

Reviewer #1: Yes

Reviewer #2: Yes

2. Has the statistical analysis been performed appropriately and rigorously? 

Reviewer #1: Yes

Reviewer #2: I Don't Know

3. Have the authors made all data underlying the findings in their manuscript fully available?

Reviewer #1: Yes

Reviewer #2: Yes

4. Is the manuscript presented in an intelligible fashion and written in standard English?

Reviewer #1: Yes

Reviewer #2: Yes

5. Review Comments to the Author

Reviewer #1: Overall it was a nicely written paper. Please see the attached document which contains my edits and comments.

A few things:

1. Please be sure to write out the entire word first before abbreviating it

2. Be sure to include which BMD site you were referring to whole body ? lumbar spine? etc.

3. Several sentences are extremely long and frankly don't make sense. Please modify.

Reviewer #2: The study investigated the effects of maternal use of TDF on renal function, infant growth, and bone related parameter in Vietnamese women, as compared to AZT. The study is important in that whether Asian women and infants are more susceptible to the detrimental impact of TDF on renal function and bone health has not been studied.

Introduction

Need to briefly introduce the use of zidovudine and if AZT has any effects on renal function and bone.

Methods

Please provide the methods of serum 25(OH) vitamin D, and other measurements, as well as the precision and accuracy of these measurements

Bone-related markers such as P1NP should have been measured.

6. PLOS authors have the option to publish the peer review history of their article (what does this mean?). If published, this will include your full peer review and any attached files.

Reviewer #1: **Yes: **Lauren Coheley

Reviewer #2: **Yes: **Jay Cao

---

## [Author Response · Author response to Decision Letter 0]

12 Apr 2021

March 31, 2021

Dr. Linglin Xie

Academic Editor

PLOS ONE

Manuscript ID: PONE-D-21-05310

Manuscript title: Influence of maternal use of tenofovir disoproxil fumarate or zidovudine in Vietnamese pregnant women with HIV on infant growth, renal function, and bone health

Authors: Kinai E, et al

Dear Dr. Xie

We were pleased to know of the positive evaluation of our manuscript and its potential acceptance for publication in PLOS ONE, subject to adequate revision and response to the reviewers' comments. 

Based on the instructions, we logged into the journal website and submitted the file of the revised manuscript (file name: PONE-D-21-05310-R1) and the file of the point-by-point response to the comments raised by the reviewers (file name: PONE-D-21-05310-response) in Microsoft Word format. 

We responded to all the comments raised by the reviewers and modified the text based on those comments. The added new text appears red in the revised manuscript.

We take this opportunity to express our gratitude to the reviewers for their constructive and useful remarks. Their comments allowed us to identify areas in our manuscript that needed modification and clarification. We also thank you for allowing us to resubmit a revised copy of the manuscript.

I hope that the revised manuscript is now acceptable for publication in PLOS ONE. 

Sincerely Yours,

Ei Kinai, MD, PhD

Department of Laboratory Medicine, Tokyo Medical University

6-7-1, Nishi-shinjuku, Shinjuku-ku, 

Tokyo, 160-0023, Japan

Phone: +81-3-3342-6111

Fax: +81-3-3340-5548

E-mail: ekinai@tokyo-med.ac.jp

Point-by-point response to the comments of Reviewer 1

Reviewer #1: 

Overall it was a nicely written paper. Please see the attached document which contains my edits and comments.

We thank the reviewer for the comments and help to improve the presentation. We replaced the original text with the text recommended by the reviewer (the revised text appears in red).

In the abstract, the reviewer modified “worse” to “lower?”. However, in the radiographic scoring system used in this study, a lower score represents better condition and a higher score reflects worse condition. In the abstract, LPV/r is associated with “higher” score, but such descriptor could be misleading and misinterpreted. Therefore, we prefer to stick to “worse” score used in the original manuscript (page 3, line 45).

A few things:

1. Please be sure to write out the entire word first before abbreviating it.

We thank the reviewer for the comment. We spell out all abbreviations, including “HIV” (abstract page 3, lines 27-28 and page 5, lines 52-3), “CD4” (page 7, line 111), and many others throughout the manuscript and Tables. The “Apgar” score refers to the physician who established the scoring system; “Virginia Apgar”, and we added the brief information on this scoring system (page 7, lines 115-117). 

2. Be sure to include which BMD site you were referring to whole body ? lumbar spine? etc.

We thank the reviewer for the comment. We specified the site of bone mineral density used in each reference. Thus, ref. 13 measured whole-body bone mineral contents, while ref.14 measured BMD of the lumbar spine (page 5-6, lines 70-75).

3. Several sentences are extremely long and frankly don't make sense. Please modify.

In response to the comment, we went through the entire text and shortened any long sentence found in the original manuscript (page 5 lines 70-74, and page 10 lines 171-172). We also corrected a misprinted sentence (page 11 lines 195-6). 

<Note to the reviewer>

We like to clarify here one issue related to the inclusion of non-HIV mother-infant pairs. As described in the manuscript, our study was designed to collect data on both HIV-infected and non-HIV mother-infant pairs. However, only two non-HIV mother-infant pairs participated in the study, and, unfortunately, neither of them visited the research center after birth. Consequently, the data of non-HIV mother-infants pairs was too small for proper comparisons with those of HIV-infected pairs. To avoid any confusion, we deleted the data of non-HIV group infants from Table 2, and deleted the text “not shown in Fig.1” (page 7, line 102, and page 10, line 174).

 

Point-by-point response to the comments of Reviewer 2

Reviewer #2: 

The study investigated the effects of maternal use of TDF on renal function, infant growth, and bone related parameter in Vietnamese women, as compared to AZT. The study is important in that whether Asian women and infants are more susceptible to the detrimental impact of TDF on renal function and bone health has not been studied.

We thank the reviewer for the positive evaluation. Since Asian women and infants are considered vulnerable to the toxicity of antiretroviral drugs due to their small body size, our study provides important information about the safety of TDF on pregnant women and infants. 

Introduction

Need to briefly introduce the use of zidovudine and if AZT has any effects on renal function and bone.

We thank the reviewer for the comment. We expanded the Introduction by adding information about the history of AZT use as perinatal prophylaxis for mother-to-child transmission of HIV (page 5, line 53-55). 

Methods

Please provide the methods of serum 25(OH) vitamin D, and other measurements, as well as the precision and accuracy of these measurements

Bone-related markers such as P1NP should have been measured.

We thank the reviewer for the comment. For measurement of serum 25 (OH) vitamin D, we used the delayed 1-step chemiluminescence microparticle immunoassay (CMIA) (Abbott ARCHITECT 25-OH Vitamin D assay). We added brief information about the method of measurement of serum 25(OH) vitamin D (page 8, lines 131-136). The ARCHITECT assay has been validated and widely used worldwide. However, we do not have the original validation data. Instead, we elected to cite one validation paper as a reference (ref. 17).

In addition to 25(OH) Vitamin D, we also expanded the text to describe the methods used for measurements of U-BMG and %TRP (page 8, lines 124-131). We also added the coefficient of variation of the methods of U-BMG employed in this study (page 8, line 125).

We agree with the reviewer that we should have measured bone-specific markers such as P1NP.

<Note to the reviewer>

We like to clarify here one issue related to the inclusion of non-HIV mother-infant pairs. As described in the manuscript, our study was designed to collect data on both HIV-infected and non-HIV mother-infant pairs. However, only two non-HIV mother-infant pairs participated in the study, and, unfortunately, neither of them visited the research center after birth. Consequently, the data of non-HIV mother-infants pairs was too small for proper comparisons with those of HIV-infected pairs. To avoid any confusion, we deleted the data of non-HIV group infants from Table 2, and deleted the text “not shown in Fig.1” (page 7, line 102, and page 10, line 174).

---

## [Editor Report · Decision Letter 1]

15 Apr 2021

Influence of maternal use of tenofovir disoproxil fumarate or zidovudine in Vietnamese pregnant women with HIV on infant growth, renal function, and bone health

PONE-D-21-05310R1

Dear Dr. Kinai,

We’re pleased to inform you that your manuscript has been judged scientifically suitable for publication and will be formally accepted for publication once it meets all outstanding technical requirements.

Kind regards,

Linglin Xie

Academic Editor

PLOS ONE
---

## [Editor Report · Acceptance letter]

19 Apr 2021

PONE-D-21-05310R1 

Influence of maternal use of tenofovir disoproxil fumarate or zidovudine in Vietnamese pregnant women with HIV on infant growth, renal function, and bone health 

Dear Dr. Kinai:

I'm pleased to inform you that your manuscript has been deemed suitable for publication in PLOS ONE. Congratulations! Your manuscript is now with our production department. 

Kind regards, 

on behalf of

Dr. Linglin Xie 

Academic Editor

PLOS ONE